# Bayesian Inference for Correlated Human Experts and Classifiers

**Markelle Kelly** [1]  **Alex Boyd** [2]  **Sam Showalter** [3]  **Mark Steyvers** [4]  **Padhraic Smyth** [1]

## Abstract

Applications of machine learning often involve making predictions based on both model outputs and the opinions of human experts. In this context, we investigate the problem of querying experts for class label predictions, using as few human queries as possible, and leveraging the class probability estimates of pre-trained classifiers. We develop a general Bayesian framework for this problem, modeling expert correlation via a joint latent representation, enabling simulation-based inference about the utility of additional expert queries, as well as inference of posterior distributions over unobserved expert labels. We apply our approach to two real-world medical classification problems, as well as to CIFAR-10H and ImageNet-16H, demonstrating substantial reductions relative to baselines in the cost of querying human experts while maintaining high prediction accuracy.

## 1. Introduction

Machine learning systems are now commonly deployed across a variety of real-world applications, including medical diagnosis, autonomous driving, and scientific discovery. In many of these applications there is significant interest in developing semi-autonomous workflows, where the predictive abilities of both human experts and machine learning models are harnessed to perform more effectively in combination than either experts or models on their own (Beck et al., 2018; Bien et al., 2018; De-Arteaga et al., 2020; Jarrett et al., 2022; Gal et al., 2022; Straitouri et al., 2023; Corvelo Benz & Rodriguez, 2024). A number of different problems have been explored in this context, including building models that "learn to defer" to human experts (Madras et al., 2018; Verma & Nalisnick, 2022) and optimally combining model

and expert predictions when both are available (Steyvers et al., 2022; Choudhary et al., 2023).

In this paper we focus on the problem of predicting the class labels that a group of human experts will produce, leveraging the output of pre-trained classifiers. Consider the following example: a radiology department at a hospital has five expert radiologists, who are responsible for classifying X-ray images of patients into one of $K$ classes. The experts do not necessarily agree on each image, given that the images are noisy and that the experts have different backgrounds and levels of expertise. Ideally, for each X-ray, the hospital would elicit class labels from all five radiologists and aggregate their predictions (e.g., via majority vote) (Owen & Grofman, 1986; Surowiecki, 2005). From the hospital's perspective this aggregate acts as ground truth, but it is too expensive to obtain on a routine basis. Now suppose that the hospital gains access to a pre-trained machine learning classifier with a minimal query cost per image. A natural question in this context is: how can the classifier be used to learn a policy that can minimize the average querying cost while accurately predicting the final majority vote?

We address this problem in a general setting for $K$-way classification with $H$ experts, where the experts produce hard labels—i.e., "votes." Examples $x$ are generated in an IID manner from some underlying unknown distribution $p(x)$. Also available per example $x$ are $K$-ary class probabilities from one or more pre-trained classifiers. Conditioned on model predictions and observed expert votes, we wish to predict the remaining unobserved expert votes, and further, to predict an aggregate function of these votes, e.g., corresponding to expert consensus or their unanimous agreement (or not). Throughout the paper we will primarily focus on the consensus function, and will use the term "consensus" to refer to the class label with the most votes (with ties broken randomly), which can include cases where the consensus vote is not a strict majority (e.g., with three labels and ten experts the consensus could be a label that gets four votes, with the other two getting three votes each).

The classifiers do not need to be trained specifically to predict the votes of the $H$ experts. In fact, our setup is particularly well-suited to situations where the classifier was trained on some labeled dataset that was generated independently from the $H$ experts, with the potential for significant

[1]Department of Computer Science, University of California, Irvine, USA [2]GE HealthCare, USA [3]Stripe, USA [4]Department of Cognitive Sciences, University of California, Irvine, USA. Correspondence to: Markelle Kelly <kmarke@uci.edu>.

*Proceedings of the $42^{nd}$ International Conference on Machine Learning*, Vancouver, Canada. PMLR 267, 2025. Copyright 2025 by the author(s).

distribution shift between the classifier predictions and the human labels and where, as a result, the relationship between the classifier probabilities and the $H$ human votes can be quite noisy.

In particular, our approach models a latent correlation structure among human experts and pre-trained classifiers. We develop a Bayesian model that jointly infers dependence among these experts and classifiers in an online manner given a sequence of examples. This allows for updating the latent model, querying the appropriate experts on a per-example basis, and inferring the predicted label of each expert. Our primary contributions are as follows.[1]

- We propose a Bayesian framework for online querying and prediction in the context of predicting the beliefs of a set of correlated human experts (Sections 3 to 6).

- We demonstrate that the proposed Bayesian approach outperforms alternative methods for this problem—using fewer expert queries on average to perfectly predict aggregate functions of expert votes—across multiple experiments with real-world image classification tasks (Section 7).

## 2. Related Work

### 2.1. Human-AI Collaboration Workflows

A wide variety of workflows for human-AI collaboration in classification tasks have been established (Green & Chen, 2019; Rastogi et al., 2023; Donahue et al., 2022), including techniques that (implicitly or explicitly) learn about specific human experts.

However, these prior approaches generally consider human-AI settings that are quite different from the problem addressed in this paper. For example, there is an extensive body of work on customizing AI for its human teammates. One family of methods involves training classifiers that are complementary to human abilities or expectations (Bansal et al., 2021a; Hemmer et al., 2022). AI-advised decision-making (Bansal et al., 2021b; Zhang et al., 2020; Liu et al., 2021; Schemmer et al., 2023), in which AI support is provided to aid a human decision-maker, can also be tailored to specific individuals (Noti & Chen, 2023; Bhatt et al., 2025). These types of workflows differ significantly from our setting, in which the goal is not to provide support to human experts, but rather to avoid querying them as much as possible by predicting their labeling decisions. Further, our approach allows for the use of a pre-trained black-box classifier, avoiding the requirement for any data about the human experts to be collected ahead of time.

Another relevant line of work focuses on the problem of assigning or delegating tasks between humans and AI. For example, classifiers can learn to defer or delegate to a human (Madras et al., 2018; Mozannar et al., 2023; Gao et al., 2023) and this can be customized for individual experts (Raman & Yee, 2021; Hemmer et al., 2023; Keswani et al., 2021; Verma et al., 2023). Other approaches model the behavior or abilities of humans and/or classifiers for the purpose of delegation (Ma et al., 2023; Fügener et al., 2021; Fuchs et al., 2022; Tudor Ionescu et al., 2016). In these settings the goal is to choose a single agent to make a decision or perform a task. In contrast, our work involves flexibly querying multiple (human and AI) agents in a sequential task-by-task manner and learning to leverage subsets of observed labels to infer unobserved labels.

### 2.2. Prediction Aggregation

Our work shares some similarities with supra-Bayesian methods, which aim to aggregate expert predictions via a posterior distribution over the true beliefs of decision makers (e.g., Winkler, 1981; Jouini & Clemen, 1996; Lindley, 1986). However, the frameworks underlying these methods differ from our setting in that they do not allow for the combination of partially observed "hard" votes (e.g., expert labels) with related probabilistic inputs (e.g., a classifier's predicted distribution over labels). Further, many of these methods assume experts are independent from one another, which can be quite restrictive in practice (Wilson, 2017). Exceptions in this context are the methods of Trick & Rothkopf (2022) and Pirš & Štrumbelj (2019), which explicitly model correlations between classifiers (but still do not allow for the incorporation of human votes).

More closely related to our work are Bayesian methods for combining human and classifier predictions. Kim & Ghahramani (2012) propose a method for combining class label votes from classifiers (or humans), modeling the relationship between the classifiers' votes and the ground truth label, including dependence between classifiers. In contrast, Steyvers et al. (2022) first query categorical confidence scores from humans, rather than a hard vote, and then integrate them with predicted classifier distributions. Finally, similar to our setting, Kerrigan et al. (2021) combine the probabilistic output of a single classifier with the single hard vote of a human expert; we will use a modified version of this method as a baseline in our experiments.

Importantly, all of these prior methods prioritize the performance of humans and/or classifiers relative to some independent ground truth. This stands apart from our setting wherein the human votes *are* considered to be the ground truth—motivated by the practical situation where we wish to use a model to infer the votes of a set of experts (e.g., radiologists) without the need to query all of them. In other

---

[1]All code and datasets used in this paper are available at https://github.com/markellekelly/consensus.

words, unlike the combination methods discussed above, our method does not combine the model and human predictions to predict some ground truth $y$, but instead uses classifier outputs and partially observed human votes to predict the remaining unobserved human votes. The goal is not to increase overall accuracy (e.g., of humans alone or classifiers alone), given that in our setting perfect accuracy is achievable by simply querying all human experts. Instead, the goal is to reduce the number of human experts that need to be queried for their predictions, while still accurately inferring what the group as a whole predicts.

The closest work that we are aware of, which predicts human expert labels in a human-AI setting, is that of Showalter et al. (2024). Their method models the consensus of human experts, incorporating the predictions of a classifier. However, their methodology differs significantly from ours in that human experts are treated as exchangeable and non-identifiable. The focus of their approach is to decide how many experts to query—not which experts to query. Likewise, their approach does not model correlations among experts, leverage information about specific experts, or predict the votes of individual experts. Our work can be seen as an extension of that of Showalter et al. (2024), in the sense that we can handle the realistic case where experts are not exchangeable, e.g., when experts are highly correlated, or vary substantially in accuracy (either overall or on a class-wise basis).

## 3. Problem Setting and Notation

Consider a stream of inputs (e.g., images) $x^{(t)} \in \mathcal{X}$ indexed sequentially by $t = 1, ..., T$, generated IID from some underlying (typically unknown) distribution $p(x)$. The task is to associate each $x^{(t)}$ with a class label $y^{(t)} \in \{1, ..., K\}$. To make this prediction, $H$ identifiable human experts are available; for any input $x^{(t)}$ each expert $i$ can be queried for their class label vote $y_i^{(t)} \in \{1, \ldots, K\}$.

The target we wish to predict is some function of these expert votes $y_*^{(t)} = f(y_1^{(t)}, ... y_H^{(t)})$. To illustrate our methodology, in the remainder of the paper we will primarily focus on consensus (as defined in Section 1) for our definition of $y_*^{(t)}$, but we note that our proposed methodology can support any function defined over expert votes. For example, decision-makers might be interested in being able to predict if *any* expert (one or more) out of the $H$ experts predicts a particular class label, e.g., a rare deadly disease in a medical setting. In Section 7.4.3 we explore the use of two such functions $f$ (other than consensus) over expert votes.

In addition to the human experts, suppose we have access to $M$ pretrained machine learning classifiers $f_i : \mathcal{X} \rightarrow$

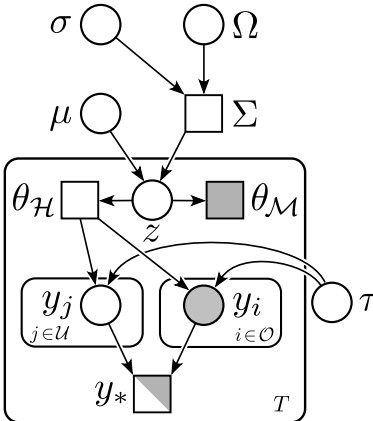

*Figure 1.* Graphical representation of the assumed generative model. Grey symbols are observed and assumed known, white are random and unobserved. Circles are random variables and rectangles are deterministic transformations of other values. Notation for hyperparameters and time-dependence $t$ is omitted for simplicity.

$\Delta^K, i = 1, ..., M$.[2] Thus, we can collect predictions from $H + M$ total agents (we will use "agents" to refer to both human experts and classifiers). We index these agents as follows: $\mathcal{M} = \{1, ..., M\}$ is the set of indices corresponding to classifier models and $\mathcal{H} = \{M+1, ..., H+M\}$ is the set of indices corresponding to human experts. Further, for any specific input $x^{(t)}$, we will use $\mathcal{O}^{(t)} \subset \mathcal{H}^{(t)}$ to denote the set of human experts whose predictions have already been observed and $\mathcal{U}^{(t)} = \mathcal{H}^{(t)} \setminus \mathcal{O}^{(t)}$ to represent experts who have not (yet) been queried, where these sets are updated as we sequentially issue queries.

For simplicity we assume that all models have zero cost and that every human expert has the same fixed cost per query. The goal is to achieve the optimal cost-accuracy trade-off—minimizing the number of expert queries given a desired level of accuracy in predicting $y_*^{(t)}$, in an online sequential fashion over classification inputs $x^{(t)}$.

## 4. A Bayesian Model for Predicting Expert Labels

We seek a model that (1) captures the relationships between the agents' predictions and (2) leverages this information to predict human expert votes (and thus consensus), conditional on any set of observed model and human predictions. To facilitate this, we model this voting process in a hierarchical Bayesian fashion using the graphical model shown in Figure 1. The outer plate (indicated by $T$) refers to the sequence of $T$ examples; nodes within this plate implicitly depend on $t$ (or, more specifically, on $x^{(t)}$), e.g., $y_i$ is really

---

[2]In our experiments we focus on the case of a single classifier (i.e., $M = 1$), but the framework is general.

$y_i^{(t)}$, where $i$ is an index over human experts. For brevity, in the remainder of the paper we suppress $t$ when it is clear from context.

Given an example $x$, each expert $i \in \mathcal{H}$ can produce a label or vote $y_i \in \{1, \ldots, K\}$ drawn from a $K$-ary categorical distribution with latent probabilities $\theta_i$ conditioned on a particular $x$. These latent expert probabilities $\theta_i$ can be interpreted as human confidence scores, similar to a predicted distribution from a classifier. Importantly, these expert probabilities are not directly observed; only associated "hard decision" votes $y_i \in \mathcal{O} \subset \mathcal{H}$ can be observed by querying. In addition, because we assume classifier queries have zero cost, predicted model probabilities $\theta_i$ for $i \in \mathcal{M}$ are assumed to be available (observed) for each example.

We transform the probability vectors to span the real domain, allowing us to leverage a multivariate normal distribution as a prior—similar to Blei & Lafferty (2006) and Pirš & Štrumbelj (2019) and building on the logistic normal model (Atchison & Shen, 1980). (An alternative approach would be an extension of the Dirichlet distribution that allows for correlations between classes and pair of agents, e.g., Trick & Rothkopf (2022)). Formally, let the additive logistic transform $\gamma : \Delta^K \to \mathbb{R}^{K-1}$ be defined as

$$\gamma(\theta) := \left[ \log \frac{\theta[1]}{\theta[K]}, \ldots, \log \frac{\theta[K-1]}{\theta[K]} \right] \quad (1)$$

where $\theta[k]$ indicates the $k^{\text{th}}$ entry in vector $\theta$. For a given probability vector $\theta_i$, latent or realized, we denote the associated transformed values as their corresponding logits: $z_i := \gamma(\theta_i)$ for $i \in \mathcal{M} \cup \mathcal{H}$. With this, we can capture correlations between classes and agents by jointly modeling both the observed model logits $z_\mathcal{M} := [z_1, \ldots, z_M]^T$ and latent human logits $z_\mathcal{H} := [z_{M+1}, \ldots, z_{M+H}]^T$ with a multivariate normal distribution

$$z := \begin{bmatrix} z_\mathcal{H} \\ z_\mathcal{M} \end{bmatrix} \sim \mathcal{N}(\mu, \Sigma) \quad (2)$$

with mean $\mu \in \mathbb{R}^d$ and covariance $\Sigma \in \mathbb{R}^{d \times d}$ for $d = (K-1)(M+H)$. Note that while each time step $t$ will have a specific set of logits $z^{(t)}$ associated with the input $x^{(t)}$, we assume they are all distributed with the same mean $\mu$ and covariance $\Sigma$. Priors are placed on these global parameters $\mu, \Sigma$, with the mean having a normal distribution, $\mu \sim \mathcal{N}(0, \sigma_\mu)$, and the covariance having a half-normal distribution on the variances, $\sigma \in \mathbb{R}_+^d \sim |\mathcal{N}|(0, \sigma_\sigma)$, and a Lewandowski-Kurowicka-Joe (LKJ) distribution (Lewandowski et al., 2009) on the correlations, $\Omega \in \mathbb{R}^{d \times d} \sim \text{LKJ}(\eta)$, where $\Sigma := \text{diag}(\sigma)\Omega$. The values $\sigma_\mu, \sigma_\sigma$, and $\eta > 0$ are all hyperparameters, which help control the model's exploration/exploitation behavior—choosing higher $\sigma_\mu$ and $\sigma_\sigma$ or $\eta > 1$ will result in higher uncertainty and thus more queries issued early-on.

While the assumed model allows for correlations within and between agent predictions, we found in our experiments that the resulting predictive distributions over human votes have the potential to be miscalibrated. To account for this, we allow for a global temperature parameter $\tau \sim |\mathcal{N}|(0, \sigma_\tau)$ to temper each latent human probability vector $\theta_i$. This results in the following distribution over votes:

$$y_i \mid \theta_i, \tau \sim \text{Categorical}(\text{TS}(\theta_i, \tau)) \text{ for } i \in \mathcal{H} \quad (3)$$

with $\text{TS}(\theta_i, \tau) := \text{softmax}(\theta_i/\tau) \propto \exp(\theta_i/\tau) \in \Delta^K$, where $1/\tau$ can be viewed as an exploration parameter (e.g., Daw et al. (2006)).

The posterior distribution for $\mu, \Sigma$, and $\tau$ (learned from the set $\mathcal{D}$ of observed agent predictions for previous examples $x^{(1)}, \ldots, x^{(t-1)}$) is intractable. In practice, we sample from this posterior distribution via MCMC sampling. In particular, at each time point $t$, we draw a set $\mathcal{S}$ of samples $\mu^s, \Sigma^s$, and $\tau^s$ (where the $s$ superscript denotes a value sampled from the posterior). Further details about the sampling procedure can be found in Appendix D.

## 5. Conditional Simulation of Expert Labels

Given $|\mathcal{S}|$ posterior samples for our underlying model parameters (based on observations up to time $t-1$), we can then perform inference about unobserved expert votes for a new input $x^{(t)}$. In particular, conditioned on the observed model predictions (logits) $z_\mathcal{M}$ and any observed human votes $y_\mathcal{O}$, we want to characterize our uncertainty about the final expert agreement $y_*^{(t)} := f(y_\mathcal{H}) \equiv f(y_\mathcal{O}, y_\mathcal{U})$ (where votes $y_\mathcal{U}$ have not yet been observed).[3]

To characterize our belief about $y_*^{(t)}$, we estimate $p(y_* = k \mid y_\mathcal{O}, z_\mathcal{M}, \mathcal{D})$ for each class $k$. It can be shown that

$$p(y_* = k \mid y_\mathcal{O}, z_\mathcal{M}, \mathcal{D}) \propto$$
$$\mathop{\mathbb{E}}_{\mu, \Sigma, \tau \mid \mathcal{D}} \left[ \mathop{\mathbb{E}}_{z_\mathcal{U}, z_\mathcal{O} \mid z_\mathcal{M}, \mu, \Sigma, \tau} \left[ \mathop{\mathbb{E}}_{y_\mathcal{U} \mid z_\mathcal{U}} \left[ \mathbb{I}(y_* = k) p(y_\mathcal{O} \mid z_\mathcal{O}) \right] \right] \right] \quad (4)$$

(see Appendix A). Note that in the case where no votes have been observed yet, with $\mathcal{O} = \emptyset$, this is simply:

$$p(y_* = k \mid z_\mathcal{M}, \mathcal{D}) \propto$$
$$\mathop{\mathbb{E}}_{\mu, \Sigma, \tau \mid \mathcal{D}} \left[ \mathop{\mathbb{E}}_{z_\mathcal{H} \mid z_\mathcal{M}, \mu, \Sigma, \tau} \left[ \mathop{\mathbb{E}}_{y_\mathcal{H} \mid z_\mathcal{H}} \left[ \mathbb{I}(y_* = k) \right] \right] \right] \quad (5)$$

Based on Equation 4, we can sample from the posterior distribution over $y_\mathcal{U}$ via Algorithm 1. (An annotated version of Algorithm 1 with additional explanation about each step

---

[3] In our description of inference and in our experiments we will generally focus on consensus, i.e., $f(y_\mathcal{H}) := \text{mode}(y_\mathcal{H})$.

can be found in Appendix B.) From these posterior samples for $y_*$ we can (for example) make a point prediction for the consensus $\hat{y}_* = \arg\max_k\{p(y_* = k \mid \ldots)\}$.

---

**Algorithm 1** Estimate $p(y_*)$ given observed predictions $y_\mathcal{O}, z_\mathcal{M}$ and the set $\mathcal{S}$ of samples from the posterior for $\mu, \Sigma$, and $\tau$

---

    `p_y_samples` $\leftarrow$ `[]` {stores $|\mathcal{S}|$ samples for $\mathbb{I}(y_* = k)p(y_\mathcal{O} \mid z_\mathcal{O})$, i.e., the inner expression in Equation 4}

    **for** `sample` $\in \mathcal{S}$ **do**

        $\mu^s, \Sigma^s, \tau^s \leftarrow$ `sample`

        **if** $|\mathcal{O}| \geq 1$ **then**

            $z_\mathcal{U}^s, z_\mathcal{O}^s \leftarrow$ draw from $p(z_\mathcal{H} \mid z_\mathcal{M})$ $\{\mathcal{N}(\mu^s, \Sigma^s)$, conditioned on $z_\mathcal{M}\}$

            $y_\mathcal{U}^s \leftarrow$ draws from $\text{Categorical}(\text{TS}(\gamma(z_i^s), \tau^s))$ for $i \in \mathcal{U}$, concatenated

            $y_*^s \leftarrow f(y_\mathcal{O}, y_\mathcal{U}^s)$

            $p(y_\mathcal{O} \mid z_\mathcal{O}^s) \leftarrow \prod_{i \in \mathcal{O}} \text{TS}(\gamma(z_i^s), \tau^s)[y_i]$

        **else**

            $z_\mathcal{U}^s \leftarrow$ draw from $p(z_\mathcal{U} \mid z_\mathcal{M})$ $\{\mathcal{N}(\mu^s, \Sigma^s)$, conditioned on $z_\mathcal{M}\}$

            $y_\mathcal{U}^s \leftarrow$ draws from $\text{Categorical}(\text{TS}(\gamma(z_i^s), \tau^s))$ for $i \in \mathcal{U}$, concatenated

            $y_*^s \leftarrow f(y_\mathcal{U}^s)$

            $p(y_\mathcal{O} \mid z_\mathcal{O}^s) \leftarrow 1$ {no observed votes}

        **end if**

        `p_y_samples.append(`$p(y_\mathcal{O} \mid z_\mathcal{O}^s) * \text{OneHot}^K[y_*^s]$`)`

    **end for**

    $p(y_* \mid y_\mathcal{O}, z_\mathcal{M}) \leftarrow$ mean of `p_y_samples`, normalized

---

## 6. Choosing Experts to Query

Now suppose, for some input $x^{(t)}$, that we have observed some set of model predictions $z_\mathcal{M}$ and $O$ human expert predictions $y_\mathcal{O}$. To determine which remaining expert $j \in \mathcal{U}$ to query next, we want to estimate how much information about the consensus $y_*$ each candidate's vote will provide. More specifically, we would like to estimate the information gain associated with observing the vote of each expert $j$. To do so, for each potential prediction $y_j' \in \{1, ..., K\}$ of expert $j$, we act as if this hypothetical prediction was actually observed, i.e., as if $y_j = y_j'$, and again draw posterior samples of $y_*$ (following Algorithm 1), including this hypothetical prediction with $y_\mathcal{O}$. We can then compute the expected entropy of the resulting $p(y_* \mid [y_\mathcal{O}, y_j], z_\mathcal{M}, \mathcal{D})$. To choose the next expert to query, we repeat this process for each candidate expert and select the expert that maximizes the information gain (or equivalently, minimizes the expected entropy), i.e., $\arg\min_{j \in \mathcal{U}}\left(\mathbb{E}_{y_j}[H(y_* | z_\mathcal{M}, [y_\mathcal{O}, y_j], \mathcal{D})]\right)$. This process is shown in detail in Algorithm 2.

To compute the expectation in the last line, we need $p(y_j \mid$

---

**Algorithm 2** Compute the expected entropy of $y_*$ after observing $y_j$

---

    `exp_entropy` $\leftarrow 0$

    $p(y_* \mid y_\mathcal{O}, z_\mathcal{M}) \leftarrow$ apply Algorithm 1 with $\mathcal{U}, \mathcal{O}, y_\mathcal{O}$

    **for** $k \in 1, ..., K$ **do**

        $\mathcal{O}' \leftarrow \mathcal{O} + \{j\}$

        $\mathcal{U}' \leftarrow \mathcal{U} \setminus \{j\}$

        $y_j' \leftarrow k$

        $p(y_* \mid [y_\mathcal{O}, y_j'], z_\mathcal{M}) \leftarrow$ apply Algorithm 1 with $\mathcal{U}', \mathcal{O}', [y_\mathcal{O}, y_j']$

        `entropy` $\leftarrow H(p(y_* \mid [y_\mathcal{O}, y_j'], z_\mathcal{M}))$

        `exp_entropy` $+= p(y_j = y_j' \mid y_\mathcal{O}, z_\mathcal{M}) *$ `entropy`

    **end for**

---

$y_\mathcal{O}, z_\mathcal{M}, \mathcal{D})$. Obtained similarly to $p(y_* = k \mid \ldots)$ in the previous section, we have

$$p(y_j = k \mid y_\mathcal{O}, z_\mathcal{M}, \mathcal{D}) \propto$$
$$\mathbb{E}_{\mu, \Sigma, \tau \mid \mathcal{D}}\left[ \mathbb{E}_{z_j, z_\mathcal{O} \mid z_\mathcal{M}, \mu, \Sigma, \tau}\left[ \mathbb{E}_{y_j \mid z_j}\left[ \mathbb{I}(y_j = k)p(y_\mathcal{O} \mid z_\mathcal{O}) \right] \right] \right].$$

After querying the selected expert, we update $\mathcal{O}, \mathcal{U}$, and $y_\mathcal{O}$ to reflect the corresponding observed vote and the process can be repeated, deciding which (if any) expert to query next. In our experiments, we halt querying for an example $x^{(t)}$ when the probability of making an error (in terms of predicting the consensus, according to the Bayesian model) falls below some error threshold $e$. If the model is well-calibrated this allows us to directly control the number of expert queries, on a per-example basis, to achieve this error—we will return to this point in Section 7.4.

## 7. Experiments

We evaluate our approach on four real-world classification tasks with corresponding classifier and human predictions, described in detail in Section 7.1. Compared to two baseline methods (described in Section 7.2), our approach consistently achieves 0% error using fewer queries on average than the baselines, as shown in Section 7.3. In Section 7.4, we highlight additional results regarding calibration, the explore-exploit trade-off, and alternative aggregation functions. Note that throughout this section "accuracy" and "error" are always defined with respect to the expert consensus, rather than some separate ground truth.

### 7.1. Datasets

Our experiments use two datasets of medical images, annotated by identifiable experts: ChestX-Ray (Nabulsi et al., 2021) and Chaoyang (Zhu et al., 2021). In addition, we include results for two datasets with simulated experts—CIFAR-10H (Peterson et al., 2019) and ImageNet-16H

(Steyvers et al., 2022). These datasets do not include identifiable experts, so we create synthetic experts with different labeling characteristics by combining the predictions of multiple non-identifiable human annotators. These datasets vary substantially in terms of the differences in performance both between individual experts and between the experts and corresponding classifier, in terms of the numbers of experts and classes, and the difficulty of the task itself—allowing us to assess our method across a variety of scenarios. We provide brief summaries of the datasets below; further details are provided in Appendix C.

**ChestX-Ray** (Wang et al., 2017) is a real-world radiology dataset released by the NIH, consisting of chest X-ray images, labeled as normal or abnormal. Nabulsi et al. (2021) collected labels from five American Board of Radiology certified radiologists for 810 of the X-ray images. These radiologists range in accuracy from 83 to 94%. We test our method using these five sets of human expert labels and the DenseNet-based classifier proposed in Tang et al. (2020), which achieves 85.7% accuracy on the test set.

**Chaoyang** (Zhu et al., 2021) is a dataset of colon images, where the task is to classify each image as either normal, serrated, adenocarcinoma, or adenoma. Three pathologists labeled each image in the 2139-image test set, with accuracies between 82 and 99%. We use the ResNet-based model described in Zhu et al. (2021) as the corresponding classifier, which attains 80.5% accuracy on the test set.

**CIFAR-10H** (Peterson et al., 2019) is an expanded version of CIFAR-10, a dataset for 10-class image classification (Krizhevsky & Hinton, 2009), which includes 50 labels from human annotators for each of the 10,000 images in the CIFAR-10 test set. CIFAR-10H annotators are not identified individually, so we modify the dataset to create synthetic experts with varying class-wise expertise. First, we combine classes to create three meta-classes (e.g., classes 1-3 become the new class 1). Then, we combine human annotations to create three experts: each expert has very high accuracy (approximately 99%) for two classes and lower accuracy (approximately 70-80%) for the remaining class, such that each expert has a different class ($k \in \{1, 2, 3\}$) of relative weakness. Across classes, the overall accuracies of these experts are between 90 and 94%. For this dataset, we use as our classifier a ResNet model with 91.5% accuracy.

**ImageNet-16H** (Steyvers et al., 2022) is similar to CIFAR-10H, comprised of human annotations for the ImageNet dataset (Deng et al., 2009), with six (non-identifiable) human labels per image. We use a modified subset of this dataset which includes three classes and three experts. Again we combine classes and human predictions to simulate experts with class-wise strengths and weaknesses; each

expert has two classes of relative expertise (about 95% accuracy), and a class of relative weakness (between 80% and 90% accuracy). Across classes, the overall accuracies of these experts are between 92 and 93%. The classifier used for this dataset is an AlexNet model with 86.0% accuracy.

## 7.2. Baseline Methods

To provide a point of comparison, we include results for two alternative approaches. Given that our approach is (to our knowledge) the first to-date to address our consensus prediction task, we modify two existing related methods (introduced in Section 2) to create appropriate baselines.

**INFEXP + $\epsilon$-greedy querying:** The original INFEXP method (Showalter et al., 2024) provides a Bayesian framework for querying experts given classifier predictions. It can be used to decide how many human experts to query to predict consensus, but has no mechanism for choosing which specific expert to query next (as experts are treated as non-identifiable). To apply this method to our setting, we exchange the random querying component of the original INFEXP method with $\epsilon$-greedy querying. To do so, we keep track of the observed accuracy (relative to consensus) of each expert. With probability $\epsilon$ a random expert is queried; otherwise, experts will be queried in order of their observed accuracy. Note that this accuracy can only be observed when enough experts are queried to determine the true consensus, e.g., if the querying stops after observing only a single expert vote, we do not update these accuracies.

**Confusion matrices + calibration:** The original confusion + calibration method (Kerrigan et al., 2021) combines model and human predictions, leveraging confusion matrices between a human expert's predictions and some ground truth. To create a baseline that can handle multiple experts and consensus prediction, we extend this approach by learning a confusion matrix for each human expert, skipping confusion matrix updates for experts whose predictions were not observed, and defining confusion with respect to the consensus rather than some separate ground truth. (As with the INFEXP + $\epsilon$-greedy baseline, this means that the model is only updated when the expert consensus is observed.) In this manner we can use this method to obtain a distribution over labels for the consensus, allowing us to predict the final majority vote and estimate the corresponding uncertainty. In experiments with this baseline, to use this method to choose which expert to query next, we apply Algorithm 2, obtaining $p(y_* \mid [y_{\mathcal{O}}, y_j'], z_{\mathcal{M}})$ via this modified confusion + calibration method rather than via our model.

## 7.3. Error vs. Cost

We generate error-cost curves by systematically varying the error threshold $e$ at which the model stops querying experts

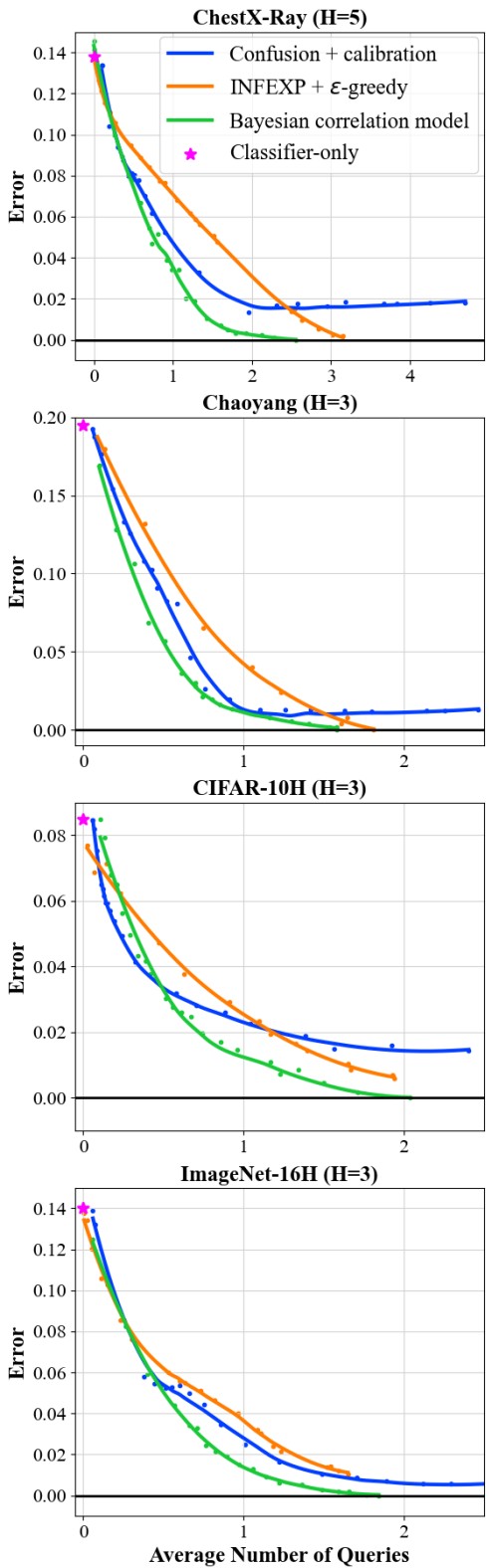

*Figure 2.* Average number of expert queries versus classification error rate for each of our four datasets.

for each example. For any specific threshold value, our model sequentially processes each example $x^{(t)}$. For each $x^{(t)}$, the model decides whether to query an expert (and, if so, which expert to query), makes that query, and repeats the process, stopping when the estimated error is below the threshold $e$. At that point, the model makes a final prediction. In addition, the set of posterior samples is updated based on the observed predictions from the model and experts, before moving to the next example. All inferences are based on classifier predictions and the partially-observed votes of any experts queried on the current or earlier examples, as described in Sections 5 and 6. Details about MCMC-based inference can be found in Section 4 and Appendix D.

We use sets of 250 examples to compute the error rate and the average querying cost for all three methods. We focus on this relatively small number of examples per run as the goal of our method is to operate in an online manner, with relatively few data points to learn from. We run each experiment with 12 different sets of examples (for datasets with less than 3000 instances, we create different sets of 250 via shuffling, since each of the methods evaluated is sensitive to the order in which data points are seen). The results reported are averaged across these 12 runs.

Figure 2 shows error-cost plots for our Bayesian method, in comparison to our baselines, for all four of our experimental datasets. These plots include the actual values observed as well as lowess-smoothed curves for each method. Note that because the curves are generated by varying the error threshold $e$ systematically (i.e., the actual error rate and number of queries are not directly controlled), the locations of specific points and distances between points vary.

For both of the real-world medical datasets (ChestX-ray and Chaoyang), our method (in green) reaches 0% error with fewer queries on average than either baseline method. In particular, while INFEXP (in orange) requires 3.16 and 1.82 queries on average to perfectly predict consensus (i.e., achieve zero error) for ChestX-ray and Chaoyang, respectively, our method requires only 2.55 and 1.58 queries. Although the confusion + calibration method (in blue) can achieve very low error rates, it never reaches 0% (i.e., perfect accuracy), even when all experts are queried. An intuition for this is that the method does not "know" it is trying to predict the consensus—so even if two out of three experts predict the same class (i.e., forming a consensus), it is possible that this method will predict a different class (e.g., in agreement with the third, dissenting expert).

We demonstrate similar results in our experiments with simulated experts. As shown in the bottom half of Figure 2, our method achieves a competitive (often the lowest) error rate across error thresholds for both CIFAR-10H and ImageNet-16H. Again, the confusion + calibration method never reaches 0% error; here, neither does INFEXP, even

with an error threshold of 0. Overall, in comparison to the baseline methods, our method consistently minimizes the error rate (i.e., reaches 0% error), and does so using the fewest queries on average.

## 7.4. Additional Results

### 7.4.1. CALIBRATION AND ERROR BOUNDING

With our approach, the stopping rule for querying is based on whether or not the model's estimated error (in predicting consensus), for each example, is below an error threshold $e$ or not; if not, the querying continues with another expert. While the classifier's own estimate of its error may be optimistically biased, we conjecture that the uncertainty estimates from our Bayesian framework will be reasonably well-calibrated. To assess model calibration, we compute expected calibration error (ECE), which compares the actual accuracy of a model with its confidence estimates (Kumar et al., 2019) (in our case, the confidence estimates are the confidence of the Bayesian model in its consensus prediction for each example $x^{(t)}$, after it has finished querying). Indeed, in our experiments, we find that our model's error estimates align well with its actual error rate, achieving low ECEs of generally less than one percent (see Table 1). Lower error thresholds tend to correspond to better ECE scores, which is what we would expect, as the model must make more queries to achieve a lower error rate, and can leverage this additional data to improve its uncertainty estimates.

| $e$ | X-Ray | Chaoyang | CIFAR | ImageNet |
|------|--------|----------|---------|----------|
| 0.05 | 0.009 | 0.008 | 0.007 | 0.009 |
| 0.025 | 0.004 | 0.003 | <0.001 | 0.003 |
| 0.01 | <0.001 | <0.001 | <0.001 | <0.001 |

*Table 1.* ECE of our method (in terms of predicting consensus) for three different error thresholds $e$ on each dataset.

As these uncertainty estimates are generally well-calibrated, setting the error threshold $e$ is a practical strategy for bounding the model's error rate. Figure 3 shows how our model's actual error rate relates to the specified error threshold. Because the model continues querying experts until its uncertainty falls below the threshold, many model predictions have an estimated error well below $e$ (e.g., if a model's uncertainty at a certain point is 0.051 but $e = 0.05$, the model will issue another query, likely reducing the uncertainty well below 0.05). Thus, as reflected in Figure 3, the model's error rate is consistently much lower than its error threshold.

### 7.4.2. EXPLORATION AND EXPLOITATION

Another advantage of our Bayesian approach is that it naturally balances exploration and exploitation in querying over time. Table 2 shows the average number of experts queried by our model in the first and last 50 examples of

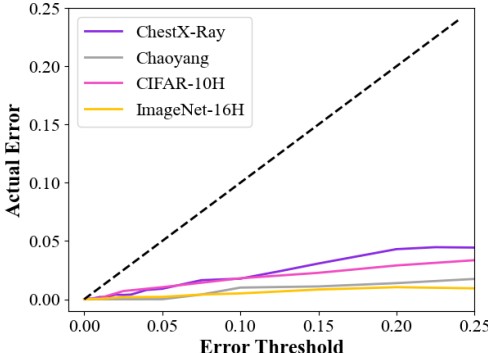

*Figure 3.* The error threshold vs. actual error rate of our model.

each set. The contrast between the two demonstrates this explore/exploit trade-off: more querying in the first 50 examples as the model learns parameters (and makes predictions), and less querying in the last 50 as it focuses on prediction.

| | X-Ray | Chaoyang | CIFAR | ImageNet |
|---|--------|----------|--------|----------|
| First 50 | 2.56 | 2.10 | 1.98 | 2.07 |
| Last 50 | 1.57 | 1.61 | 1.10 | 1.32 |

*Table 2.* Average number of experts queried by our model for the first and last 50 examples (out of 250 examples in total) using an error threshold of $e = 0.01$.

Further, because our method operates in an online fashion, it is especially important that it can handle distribution shifts in the input data. In the context of our motivating radiology example, for instance, the distribution of X-ray image data could suddenly shift if the X-ray machine is replaced with a newer model or its operating parameters are changed. To investigate this scenario, we create a version of the ImageNet-16H dataset with such a shift. ImageNet-16H includes images with varying levels of noise (Steyvers et al., 2022); we create 48 test sets of 250 images where the first 125 are low-noise and the next 125 are high-noise. To adapt our model to this setting, we use a "sliding window" of size 50, i.e., using only the 50 most recent examples in making predictions (see further details in Appendix D).

Figure 4 shows the number of queries made by our model over time. The first half of this plot demonstrates the exploration/exploitation strategy we expect: the model makes more queries on average when it has seen very few data points, using fewer and fewer queries as time progresses. However, at the halfway point, when classifier and expert accuracies drop (due to the shift to noisier images), the average number of queries immediately increases. Again, as the model adapts to the new distribution, the number of queries falls—although it remains higher than in the easier, low-noise first half. By increasing exploration after this major shift in the input data, our Bayesian model maintains a 0%

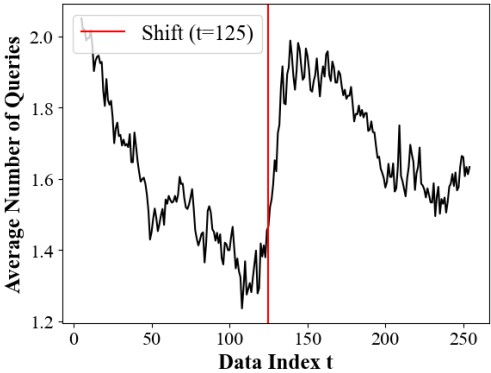

*Figure 4.* The average number of queries issued by our model over time (averaged over 48 experimental runs) for a modified version of ImageNet-16H with a distribution shift at t=125.

error rate despite the experts' drops in accuracy.

### 7.4.3. ALTERNATIVE AGGREGATION FUNCTIONS

Although we have focused on predicting majority vote, our method models the belief of each expert individually, enabling the prediction of any function over the expert votes $f(y_1, ..., y_H)$. In this section we investigate the use of our model with two practical alternative aggregation functions.

In the case of predicting the presence of a rare disease, even one positive prediction from a single radiologist may be sufficient reason to pursue further testing (Adachi et al., 2025). Formally, in the case where $y_i \in \{0, 1\}$, we can use the aggregation function $f_{\text{any}}(y_1, ..., y_H) = \mathbb{I}\left[\sum_{i=1}^{H} y_i \geq 1\right]$. Alternatively, we may want to give a positive diagnosis only if the experts are unanimously positive, i.e., aggregating with $f_{\text{all}}(y_1, ..., y_H) = \prod_{i=1}^{H} y_i$.

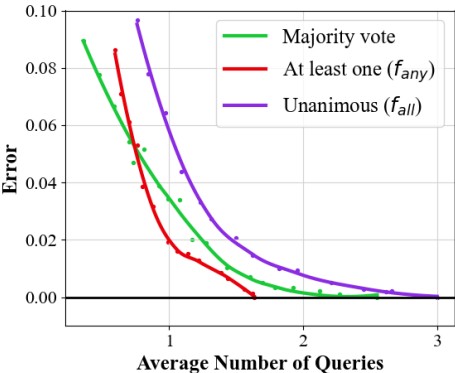

*Figure 5.* Error vs. cost for the ChestX-ray dataset in predicting majority vote (green), in comparison to predicting whether any expert will vote 1 (red) and whether the vote will be unanimous (purple).

Error-cost curves for the ChestX-ray dataset, using these

aggregation functions, are shown in Figure 5. Compared to the original curve, the error-cost curve for $f_{\text{any}}$ starts with a higher error rate for the same number of queries (on average), but the curve drops off more steeply, reflecting that a single query can be much more informative in this case than in the majority vote case. In contrast, when using $f_{\text{all}}$, the model makes more queries on average—demonstrating that, on average, more expert votes are needed in this setting.

## 8. Discussion and Conclusion

**Limitations**  There are a number of potential limitations to our overall approach and experiments. First, our evaluation is limited to image classification tasks; our findings need not necessarily generalize to other settings, e.g., using text or tabular data. Further, for computational and estimation efficiency, our experiments focused on settings with relatively small numbers of class labels $K$ and numbers of agents $(M + H)$, given that the underlying covariance matrices $\Sigma$ grow quickly with $K$, $M$, and $H$. For larger values of $K$, a more structured model for $\Sigma$ (e.g., low-rank approximations, informative priors) may be useful and is an interesting avenue for further research.

**Future Work**  As our model is relatively flexible, a number of interesting extensions are worth exploring. One such extension would be to allow human experts to provide verbal levels of confidence for their predictions (e.g., as in Steyvers et al. (2022)), which then could be associated with latent confidence scores. Another potentially useful direction for future work, particularly in real-world settings, is the incorporation of different query costs for different agents.

**Summary**  In this paper we develop a systematic Bayesian approach to address the under-explored problem of making joint inferences about expert labels in the context of $K$-ary classification, conditioned on partial information. The method is intuitive and flexible and it can be readily adapted for various settings (for example, using the sliding-window approach from the previous section to handle distribution shifts, or by simulating the consensus of only a subset of experts in the case that some experts become unavailable). Further, as shown in our experiments, this method consistently achieves error rates below a specified error threshold, and minimizes error with lower average querying costs than those of baseline methods. Altogether, these findings demonstrate the potential of our methodology for streamlining prediction in real-world human-AI classification tasks.

## Acknowledgements

We would like to acknowledge the specific and constructive feedback from the reviewers and Area Chair, which improved the quality of the final paper. This research was

supported in part by the National Science Foundation under awards RI-1900644 and RI-1927245 and in part by the Hasso Plattner Institute (HPI) Research Center in Machine Learning and Data Science at the University of California, Irvine.

## Impact Statement

Our method leverages the outputs of a classifier to predict the class labels provided by human experts, while reducing the costs of consulting human experts. This has the potential for positive societal impact, by improving accuracy in terms of the expert consensus when it is not practical to query many human experts, or by saving the time and money that would have been required to query each individual expert. However, while our method is designed to reflect expert agreement, it can make a prediction for a particular input without querying any experts. Thus, while this methodology may be useful for scenarios in which human expertise is critical, it could also result in misplaced confidence that human experts were involved in a particular class prediction, and is not directly applicable to high-stakes settings where it is necessary to have a human in-the-loop. Ultimately, responsible deployment with human oversight is crucial for this methodology to be used safely and effectively.

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

## A. Derivations of Inference Equations

Below we assume that all expressions are implicitly conditioned on $\mathcal{S}$ (and reintroduce it notationally at the end of this section). For each class $k$ we wish to compute

$$p(y_* = k \mid y_\mathcal{O}, z_\mathcal{M}) = \int_{y_\mathcal{U}} p(y_* = k, y_\mathcal{U} \mid y_\mathcal{O}, z_\mathcal{M}) dy_\mathcal{U}$$

$$= \int_{y_\mathcal{U}} p(y_* = k \mid y_\mathcal{U}, y_\mathcal{O}, z_\mathcal{M}) p(y_\mathcal{U} \mid y_\mathcal{O}, z_\mathcal{M}) dy_\mathcal{U}$$

Since $y_* = \mathrm{mode}(y_\mathcal{U}, y_\mathcal{O})$ (i.e., is a deterministic function of $y_\mathcal{U}$ and $y_\mathcal{O}$) we have:

$$= \int_{y_\mathcal{U}} \mathbb{I}(y_* = k) p(y_\mathcal{U} \mid y_\mathcal{O}, z_\mathcal{M}) dy_\mathcal{U} \tag{6}$$

Focusing on the latter probability we can obtain:

$$p(y_\mathcal{U} \mid y_\mathcal{O}, z_\mathcal{M})$$

$$= \int_{z_\mathcal{U}} p(y_\mathcal{U}, z_\mathcal{U} \mid y_\mathcal{O}, z_\mathcal{M}) dz_\mathcal{U}$$

$$= \int_{z_\mathcal{U}} p(y_\mathcal{U} \mid z_\mathcal{U}, y_\mathcal{O}, z_\mathcal{M}) p(z_\mathcal{U} \mid y_\mathcal{O}, z_\mathcal{M}) dz_\mathcal{U}$$

$$= \int_{z_\mathcal{U}} p(y_\mathcal{U} \mid z_\mathcal{U}) p(z_\mathcal{U} \mid y_\mathcal{O}, z_\mathcal{M}) dz_\mathcal{U} \tag{7}$$

Narrowing our focus again, we can manipulate the second probability in 7:

$$p(z_\mathcal{U} \mid y_\mathcal{O}, z_\mathcal{M})$$

$$= \int_{z_\mathcal{O}} p(z_\mathcal{U}, z_\mathcal{O} \mid y_\mathcal{O}, z_\mathcal{M}) dz_\mathcal{O}$$

$$= \int_{z_\mathcal{O}} \frac{p(y_\mathcal{O}, z_\mathcal{U}, z_\mathcal{O} \mid z_\mathcal{M})}{p(y_\mathcal{O} \mid z_\mathcal{M})} dz_\mathcal{O}$$

$$= \frac{\int_{z_\mathcal{O}} p(y_\mathcal{O} \mid z_\mathcal{U}, z_\mathcal{O}, z_\mathcal{M}) p(z_\mathcal{U}, z_\mathcal{O} \mid z_\mathcal{M}) dz_\mathcal{O}}{p(y_\mathcal{O} \mid z_\mathcal{M})}$$

$$= \frac{\int_{z_\mathcal{O}} p(y_\mathcal{O} \mid z_\mathcal{O}) p(z_\mathcal{U}, z_\mathcal{O} \mid z_\mathcal{M}) dz_\mathcal{O}}{\int_{z_\mathcal{O}} p(y_\mathcal{O} \mid z_\mathcal{O}) p(z_\mathcal{O} \mid z_\mathcal{M}) dz_\mathcal{O}} \tag{8}$$

Reincorporating 8 into 7 we obtain:

$$p(y_\mathcal{U} \mid y_\mathcal{O}, z_\mathcal{M})$$

$$= \int_{z_\mathcal{U}} p(y_\mathcal{U} \mid z_\mathcal{U}) \frac{\int_{z_\mathcal{O}} p(y_\mathcal{O} \mid z_\mathcal{O}) p(z_\mathcal{U}, z_\mathcal{O} \mid z_\mathcal{M}) dz_\mathcal{O}}{\int_{z_\mathcal{O}} p(y_\mathcal{O} \mid z_\mathcal{O}) p(z_\mathcal{O} \mid z_\mathcal{M}) dz_\mathcal{O}} dz_\mathcal{U}$$

$$= \frac{\int_{z_\mathcal{U}} \int_{z_\mathcal{O}} p(y_\mathcal{U} \mid z_\mathcal{U}) p(y_\mathcal{O} \mid z_\mathcal{O}) p(z_\mathcal{U}, z_\mathcal{O} \mid z_\mathcal{M}) dz_\mathcal{O} dz_\mathcal{U}}{\int_{z_\mathcal{O}} p(y_\mathcal{O} \mid z_\mathcal{O}) p(z_\mathcal{O} \mid z_\mathcal{M}) dz_\mathcal{O}} \tag{9}$$

Reincorporating 9 into 6 we have:

$$
p(y_* = k \mid y_\mathcal{O}, z_\mathcal{M})
$$

$$
= \int_{y_\mathcal{U}} \mathbb{I}(y_* = k) \frac{\int_{z_\mathcal{U}} \int_{z_\mathcal{O}} p(y_\mathcal{U} \mid z_\mathcal{U}) p(y_\mathcal{O} \mid z_\mathcal{O}) p(z_\mathcal{U}, z_\mathcal{O} \mid z_\mathcal{M}) dz_\mathcal{O} dz_\mathcal{U}}{\int_{z_\mathcal{O}} p(y_\mathcal{O} \mid z_\mathcal{O}) p(z_\mathcal{O} \mid z_\mathcal{M}) dz_\mathcal{O}} dy_\mathcal{U}
$$

$$
\propto \int_{y_\mathcal{U}} \mathbb{I}(y_* = k) \int_{z_\mathcal{U}} \int_{z_\mathcal{O}} p(y_\mathcal{U} \mid z_\mathcal{U}) p(y_\mathcal{O} \mid z_\mathcal{O}) p(z_\mathcal{U}, z_\mathcal{O} \mid z_\mathcal{M}) dz_\mathcal{O} dz_\mathcal{U} dy_\mathcal{U}
$$

$$
= \int_{z_\mathcal{U}} \int_{z_\mathcal{O}} \int_{y_\mathcal{U}} \mathbb{I}(y_* = k) p(y_\mathcal{U} \mid z_\mathcal{U}) p(y_\mathcal{O} \mid z_\mathcal{O}) p(z_\mathcal{U}, z_\mathcal{O} \mid z_\mathcal{M}) dy_\mathcal{U} dz_\mathcal{O} dz_\mathcal{U}
$$

$$
= \mathop{\mathbb{E}}_{z_\mathcal{U}, z_\mathcal{O} \mid z_\mathcal{M}} \left[ \mathop{\mathbb{E}}_{y_\mathcal{U} \mid z_\mathcal{U}} \left[ \mathbb{I}(y_* = k) p(y_\mathcal{O} \mid z_\mathcal{O}) \right] \right]
$$

Ultimately we have:

$$
p(y_* = k \mid y_\mathcal{O}, z_\mathcal{M}, \mathcal{S})
$$

$$
\propto \mathop{\mathbb{E}}_{\mu, \Sigma, \tau \mid \mathcal{S}} \left[ \mathop{\mathbb{E}}_{z_\mathcal{U}, z_\mathcal{O} \mid z_\mathcal{M}, \mu, \Sigma, \tau} \left[ \mathop{\mathbb{E}}_{y_\mathcal{U} \mid z_\mathcal{U}} \left[ \mathbb{I}(y_* = k) p(y_\mathcal{O} \mid z_\mathcal{O}) \right] \right] \right] \tag{10}
$$

## B. Algorithm 1 with Annotations

---

**Algorithm 3** Estimate $p(y_*)$ given observed predictions $y_\mathcal{O}, z_\mathcal{M}$ and the set $\mathcal{S}$ of samples from the posterior for $\mu, \Sigma$, and $\tau$

---

p_y_samples $\leftarrow$ []
**for** sample $\in \mathcal{S}$ **do**

$\mu^s, \Sigma^s, \tau^s \leftarrow$ sample {*These correspond to a single sample from the posterior distribution. This loop will draw a single, corresponding sample for $z_\mathcal{U}$ and $z_\mathcal{O}$, followed by a sample for $y_\mathcal{U}$, giving us a complete, joint sample from our model over agent predictions.*}

**if** $|\mathcal{O}| \geq 1$ **then**

$z_\mathcal{U}^s, z_\mathcal{O}^s \leftarrow$ draw from $p(z_\mathcal{H} \mid z_\mathcal{M})$ {*If any human predictions have been observed, we will need to re-weigh by $p(y_\mathcal{O} \mid z_\mathcal{O})$, which we compute in this block. The first step is to sample $z_\mathcal{H}$, which is drawn from $\mathcal{N}(\mu^s, \Sigma^s)$, conditioned on $z_\mathcal{M}$ (using the closed-form conditional multivariate normal distribution).*}

$y_\mathcal{U}^s \leftarrow$ draws from Categorical$(\text{TS}(\gamma(z_i^s), \tau^s))$ for $i \in \mathcal{U}$, concatenated {*In this step, we are drawing a sample for each unobserved expert vote $y_i$, from the corresponding categorical distribution for expert $i$.*}

$y_*^s \leftarrow f(y_\mathcal{O}, y_\mathcal{U}^s)$ {*Here, we compute the final output of the aggregation function $f$ on all expert votes: both the observed expert votes $y_\mathcal{O}$ and the sampled, unobserved expert votes $y_\mathcal{U}^s$. This sampled result $y_*^s$ is needed to compute the corresponding entry of* p_y_samples *(see below).*}

$p(y_\mathcal{O} \mid z_\mathcal{O}^s) \leftarrow \prod_{i \in \mathcal{O}} \text{TS}(\gamma(z_i^s), \tau^s)[y_i]$ {*This step computes $p(y_\mathcal{O} \mid z_\mathcal{O})$ for the sample $z_\mathcal{O}^s$. Since the $y_i$s are conditionally independent given $z_\mathcal{O}$, this is the product of each $p(y_i|z_i)$, for $i \in \mathcal{O}$. The inside of the product is the $y_i^{th}$ element of the corresponding categorical distribution, i.e., the probability of observing the class label $y_i$ under our sampled categorical distribution $\text{TS}(\gamma(z_i^s), \tau^s)$.*}

**else**

$z_\mathcal{U}^s \leftarrow$ draw from $p(z_\mathcal{U} \mid z_\mathcal{M})$ {$\mathcal{N}(\mu^s, \Sigma^s)$, *conditioned on $z_\mathcal{M}$*} {*In this if-block, we cover the case in which we have not yet observed any experts. Similarly to above, we sample $z_\mathcal{H}$ (which is equal to $z_\mathcal{U}$), drawing from $\mathcal{N}(\mu^s, \Sigma^s)$, conditioned on $z_\mathcal{M}$.*}

$y_\mathcal{U}^s \leftarrow$ draws from Categorical$(\text{TS}(\gamma(z_i^s), \tau^s))$ for $i \in \mathcal{U}$, concatenated {*We draw a sample for each (unobserved) expert vote $y_i$, from the corresponding categorical distribution for expert $i$.*}

$y_*^s \leftarrow f(y_\mathcal{U}^s)$ {*We compute the final output of the aggregation function $f$ on these sampled expert votes. This sampled result $y_*^s$ is needed to compute the corresponding entry of* p_y_samples *(see below).*}

$p(y_\mathcal{O} \mid z_\mathcal{O}^s) \leftarrow 1$ {*Since we have not observed any expert votes, the sampled result $y_*^s$ is not re-weighted.*}

**end if**

p_y_samples.append($p(y_\mathcal{O} \mid z_\mathcal{O}^s) * \text{OneHot}^K[y_*^s]$) { *The array* p_y_samples *contains samples of the innermost part of the expectation in Equation 4, i.e., for each sample, we add a $K$-dimensional vector to* p_y_samples*, where the element corresponding to the sampled expert agreement $y_*^s$ is $p(y_\mathcal{O} \mid z_\mathcal{O}^s)$ (in the case where no expert votes have been observed, this is 1) and the rest of the elements are 0. This corresponds to $|S|$ samples of the expert agreement, re-weighted based on the expert votes that have already been observed.*}

**end for**

$p(y_* \mid y_\mathcal{O}, z_\mathcal{M}) \leftarrow$ mean of p_y_samples, normalized {*In this final step, we take the normalized mean of* p_y_samples *to obtain a probability distribution over the agreement $y_*$.*}

---

## C. Additional Details on Datasets

**ChestX-ray**   Here we report the accuracies of all five experts and the classifier, a DenseNet-121 convolutional neural network trained for this task (see (Tang et al., 2020) for model details). Note that we treat this as a binary classification task (normal vs. abnormal).

|  | Overall |
|---|---|
| Classifier | 86.2% |
| Expert 1 | 90.5% |
| Expert 2 | 92.3% |
| Expert 3 | 83.7% |
| Expert 4 | 93.2% |
| Expert 5 | 88.3% |

*Table 3.* Overall accuracies for ChestX-ray classifier and experts

**Chaoyang**   Here we report the accuracies of the three experts and the classifier, a ResNet model trained for this task (see (Zhu et al., 2021) for model details).

|  | Class 1 | Class 2 | Class 3 | Class 4 | Overall |
|---|---|---|---|---|---|
| Classifier | 91.2% | 55.3% | 96.8% | 67.2% | 80.5% |
| Expert 1 | 67.5% | 78.8% | 99.2% | 96.3% | 85.9% |
| Expert 2 | 92.3% | 62.9% | 98.4% | 59.9% | 81.7% |
| Expert 3 | 100.0% | 96.5% | 100.0% | 100.0% | 99.2% |

*Table 4.* Class-wise and overall accuracies for Chaoyang classifier and experts

**CIFAR-10H**   We combine classes so that the new class 1 corresponds to the original classes 1, 2, and 3; class 2 corresponds to classes 4, 5, and 6; and class 3 corresponds to classes 7, 8, 9, and 10. We then create the combined experts as follows. The predictions of expert 1 are taken from the first 15 columns of annotator labels; for classes 1 and 3 their prediction is the consensus of the first 10 of these labels; for class 2 their prediction is a "minority opinion" of all 15 labels (a random prediction that is not part of the consensus, if one exists). Expert 2 is formed similarly from the next 16 labels; the majority vote of the first 10 for classes 1 and 2 and a minority opinion from all 15 for class 3. Expert 3 is formed in the same way using the following 15 columns, with the majority vote for classes 2 and 3, and the minority opinion for class 1. The class-wise and overall performance of these experts and the classifier are shown in Table 5.

|  | Class 1 | Class 2 | Class 3 | Overall |
|---|---|---|---|---|
| Classifier | 82.0% | 95.6% | 95.6% | 91.5% |
| Expert 1 | 100.0% | 78.9% | 99.8% | 93.7% |
| Expert 2 | 99.7% | 99.7% | 75.5% | 89.8% |
| Expert 3 | 69.5% | 99.6% | 99.9% | 99.6% |

*Table 5.* Class-wise and overall accuracies for CIFAR-10H classifier (ResNet) and experts

**ImageNet-16H**   We combine classes as follows. The new class 1 corresponds to the original classes for "clock," "knife," "oven," "chair," "bottle," "keyboard" (roughly: "objects"). The new class 2 combines the original classes "cat," "elephant," "dog," "bird," "bear" ("animals"). The new class 3 combines "airplane," "boat," "car," "truck," "bicycle" ("transportation"). Experts are then combined following the same strategy as described for CIFAR-10H, with expert 1 using predictions from the first 67 annotation columns, expert 2, the next 66, and expert 3, the last 67. As there are multiple predictions per image depending on the noise level, we choose annotations based on noise level to further diversify the experts: experts 1 and 3 use predictions for $\Omega$=95 and 110, while expert 2 uses predictions for $\Omega$=110 and 125. The final class-wise and overall accuracies are shown in 8.

|  | Class 1 | Class 2 | Class 3 | Overall |
|---|---|---|---|---|
| Classifier | 81.1% | 95.9% | 82.4% | 86.0% |
| Expert 1 | 96.6% | 82.4% | 97.9 % | 92.6% |
| Expert 2 | 95.5% | 98.1% | 81.2% | 91.7% |
| Expert 3 | 86.9% | 95.9% | 96.0% | 92.5% |

*Table 6.* Class-wise and overall accuracies for ImageNet-16H classifier (AlexNet) and experts

## D. Experimental Details

### Modeling

Experiments were run on an NVIDIA GeForce 2080ti GPU over the course of several days. The following hyperparameter values were used for all experiments:

$$\sigma_T = 0.4$$
$$\sigma_\mu = 0.1$$
$$\eta = 0.75$$
$$\sigma_\sigma = 1$$

Parameters were updated after every example for the first 20 examples, then every 10 examples until 100 total examples were reached, at which point the update rate was further reduced to every 50 examples.

For all inference tasks (for example, to estimate the parameters at each time step $t$), we used three independent Markov chains, each comprising 1,500 warm-up iterations followed by 2,000 post-warm-up (posterior) samples. The resulting 6,000 samples were combined into our final sampled approximate posterior distribution. Chain convergence plots generally indicated that chains mixed; R-hat values across parameters and experiments were consistently within +/-.001 of 1.

### Distribution Shift Experiment

We include here the performance of the classifier and each expert before and after the distribution shift. Both overall per-agent performance and inter-agent, inter-class differences change substantially with the distribution shift.

|  | Class 1 | Class 2 | Class 3 | Overall |
|---|---|---|---|---|
| Classifier | 85.3% | 100.0% | 88.8% | 87.0% |
| Expert 1 | 100.0% | 100.0% | 100.0% | 100.0% |
| Expert 2 | 100.0% | 100.0% | 86.0% | 93.3% |
| Expert 3 | 80.7% | 100.0% | 100.0% | 89.9% |

*Table 7.* Class-wise and overall accuracies for ImageNet-16H classifier and experts, "low noise" ($\Omega$=80) examples

|  | Class 1 | Class 2 | Class 3 | Overall |
|---|---|---|---|---|
| Classifier | 78.8% | 44.4% | 82.4% | 81.6% |
| Expert 1 | 100.0% | 44.4% | 100.0% | 98.7% |
| Expert 2 | 99.5% | 100.0% | 57.7% | 81.3% |
| Expert 3 | 35.0% | 88.8% | 100.0% | 64.5% |

*Table 8.* Class-wise and overall accuracies for ImageNet-16H classifier and experts, "high noise" ($\Omega$=125) examples

These experiments were run with a "sliding window," done by re-training the model after each new data point and passing in only the most recent 50 observed points. The model was given an error threshold of $e = 0.01$.

