# OpenReview forum: "Bayesian Inference for Correlated Human Experts and Classifiers"
_ICML.cc/2025/Conference — ICML 2025 poster_

### Official Review · Reviewer_YjoC · 2025-03-13

**Overall Recommendation:** 2

**Summary:**

This paper addresses the challenge of predicting human expert consensus in $K$-ary classification by developing a Bayesian framework that leverages model outputs and expert queries. Given a limited budget for expert input, the method efficiently maximizes prediction accuracy by modeling expert correlation and inferring unobserved labels. Experiments on medical and image datasets demonstrate significant cost reductions while maintaining accuracy.

------post rebuttal

The rebuttal clarified a few points for me, and I now have a better appreciation of the motivation. The authors claim that the method can be extended to incorporate varying expert costs, but since they do not provide detailed derivations, it is difficult to verify this during the rebuttal phase. I also find that the Bayesian method adopted in the paper is fairly standard, and I would have liked to see more advanced Bayesian techniques. Therefore, I maintain my score.

**Claims And Evidence:**

Yes.

**Essential References Not Discussed:**

N/A.

**Experimental Designs Or Analyses:**

Yes.

**Methods And Evaluation Criteria:**

Yes.

**Other Comments Or Suggestions:**

There is no conclusion provided.

**Other Strengths And Weaknesses:**

I have concerns regarding the motivation for Consensus Prediction. Specifically, given the assumption that pre-trained classifiers are provided, it is unclear why expert consultation is necessary for prediction. Furthermore, it raises the question of why pre-trained classifiers cannot themselves be considered experts.

Furthermore, after framing this problem, the Bayesian framework is rather natural, and I do not consider the proposed methodology innovative and significant.

Also, they assume all experts cost the same, which isn't very practical.

**Questions For Authors:**

1. Could the authors elaborate on the motivation for the problem, and the innovation and significance of the proposed methodology?
2. Could the methodology be extended to incorporate varying expert costs, rather than assuming uniform costs?

**Relation To Broader Scientific Literature:**

The discussion of related work in Section 2 is properly written.

**Theoretical Claims:**

There are no proofs provided.

---

### Official Review · Reviewer_qbrk · 2025-03-13

**Overall Recommendation:** 3

**Summary:**

The paper proposes a hierarchical Bayesian model for aggregating predictions from  pretrained classifiers and  human experts, aiming to estimate the majority vote outcome. It assumes the ground truth corresponds to the majority vote of human experts, with each expert's latent probability correlated and modeled using a multivariate normal distribution. This distribution employs hyperpriors: the mean follows a normal distribution, and the covariance is modeled using the Lewandowski-Kurowicka-Joe distribution. The proposed model not only predicts the estimated majority vote results but also strategically determines which expert to query next by maximizing information gain, equivalently minimizing the expected entropy. Empirical results demonstrate that the Bayesian model proposed outperforms existing aggregation methods.

## Update after rebuttal
I found the authors’ rebuttal satisfactory. Their proposed revisions addressed my main concerns—namely, the general applicability of the social welfare function for consensus and the motivating example for the new setting. Accordingly, I have increased my score to 3.

**Claims And Evidence:**

The proposed method is based on a Bayesian framework, and its effectiveness is empirically validated on four datasets against two baseline methods. The results clearly demonstrate that the proposed method consistently outperforms both baselines across all datasets in terms of the trade-off between the average number of expert queries and classification error. Additionally, the authors claim that their Bayesian model is well-calibrated, as evidenced by low expected calibration error (ECE) metrics in Table 1 and Figure 4.

**Essential References Not Discussed:**

The paper appears to cover essential references adequately; however, it remains possible that there are important related works not mentioned, particularly those outside my familiarity.

**Experimental Designs Or Analyses:**

I did not verify the implementation details explicitly; however, the choice of datasets and the use of the NUTS sampler appear reasonable.

**Methods And Evaluation Criteria:**

Overall, the adoption of a hierarchical Bayesian approach is well-motivated for the problem at hand, especially given the inherent uncertainty and correlation among human experts. The evaluation approach, focusing on the trade-off between classification accuracy and the number of expert queries, is appropriate and clearly demonstrates the cost-effectiveness of the proposed method. Additionally, the emphasis on model calibration through expected calibration error (ECE) metrics further strengthens the validity of the approach.

**Other Comments Or Suggestions:**

The paper does not provide an explanation of what the Expected Calibration Error (ECE) represents.

**Other Strengths And Weaknesses:**

Strength
- The manuscript is well-written.
- Figure 1 helps significantly in understanding the overall pipeline.

Weakness
- The proposed approach appears rather straightforward and incremental. Indeed, the authors themselves acknowledge in Section 2.2 that this paper extends the work of Showalter et al. (2024) by incorporating correlations among experts' predictions. Consequently, the primary novelty lies in modeling the latent probability $z$ using a hyperprior. However, I find this hierarchical Bayesian approach to be quite standard practice within the community. While there is certainly nothing problematic about utilizing established methods, for a submission to a top-tier conference like ICML, I would expect a more innovative methodological contribution. Therefore, I suggest the authors consider submitting their paper to a journal such as Knowledge-Based Systems, where incremental methodological advancements combined with practical significance are well-received. For Bayesian experimental design part, a similar setting has been investigated in [1], albeit with slight differences.
[1] Bayesian Optimization for Building Social-Influence-Free Consensus, https://doi.org/10.48550/arXiv.2502.07166
- This paper appears to conflate the concept of consensus with majority voting. It is important to clarify that Bayesian consensus is not inherently restricted to a utilitarian (majority-based) approach. I am not entirely convinced that utilitarian aggregation is universally appropriate, especially in high-stakes contexts requiring careful deliberation, such as X-ray classification tasks. In the context of medical diagnostics like X-ray analysis, human expert labels should be treated as informed advice rather than definitive ground truths. Definitive ground truths in such scenarios are typically only attainable via invasive procedures, such as surgery, which are prohibitively costly. Hence, expert labels serve as practical proxies. The primary motivation behind aggregating multiple expert opinions is to enhance the reliability and robustness of these proxy judgments. Adopting a majority-vote strategy aligns with the "wisdom of crowds" principle. However, an egalitarian approach may be more suitable in situations where avoiding the risk of overlooking critical indicators is paramount. Given that individual doctors possess varied experiences and specialized expertise, aggregating multiple expert assessments serves primarily to mitigate the risk of missing rare but significant findings. In this egalitarian scenario, if any expert identifies an abnormality, the consensus decision should categorize the image as abnormal, independent of the majority opinion. Fortunately, the approach described in this paper seems sufficiently flexible to accommodate both utilitarian and egalitarian aggregation schemes. I encourage the authors to further develop their methodology to explicitly support a general framework, allowing practitioners to select the appropriate social-welfare functional based on their specific decision-making needs.

**Questions For Authors:**

See weakness.

**Relation To Broader Scientific Literature:**

Human-AI collaboration has become increasingly important, especially in light of the rapid emergence of large language models (LLMs). Aligning AI models with multiple human preferences is an essential topic in the broader scientific literature. However, this paper assumes a fixed and identifiable set of human experts. A more general and practically relevant setting would relax this assumption, considering scenarios where expert identities are neither fixed nor necessarily known.

**Theoretical Claims:**

Not applicable; The paper does not claim theoretical results.

---

### Official Review · Reviewer_g7pw · 2025-03-13

**Overall Recommendation:** 3

**Summary:**

This paper introduces a Bayesian framework for predicting the consensus of human experts in K-ary classification tasks, leveraging pre-trained classifiers to minimize the cost of querying experts while maintaining high accuracy. The correlation between human experts and classifiers was modelled by a joint latent representation to infer posterior distribution over unobserved expert labels. Results on medical datasets and image benchmark datasets CIFAR, ImageNet were reported.

**Claims And Evidence:**

Yes, the claim was basically supported by results in Section 7.3 and 7.4. But it seems the authors make a specific niche setting under this topic, and no other existing algorithms can be directly used to be compared with their method. They modify two existing methods as benchmarks, but it is hard to know whether the benchmarks are suitable.

**Essential References Not Discussed:**

Not sure, I am not an expert in this specific field.

**Experimental Designs Or Analyses:**

Seems lacking experiments about sensitivity of MCMC hyperparameters to the final performance.

Seems also lacking evaluation of how different pretrained ML classifiers affect the final results.

**Methods And Evaluation Criteria:**

The proposed method tries to predict expert consensus while minimising the query number. The zero error point is used as an important condition for evaluation, which seems reasonable.

The proposed method adopts MCMC for sampling and posterior computing. It is common that MCMC might take a long time to burn-in or fail to estimate the correct posterior region in the defined search space. But I cannot see any ablation studies or results related to how MCMC hyper-parameters may affect overall performance, as well as the failure rate (if any) of MCMC in this problem etc.

**Other Comments Or Suggestions:**

* Might be good to further compress the introduction of datasets, - a bit too long in the main content of a conference paper, leaving less space for more important parts.

**Other Strengths And Weaknesses:**

Strengths:

* Well structured paper, easy to follow
* The proposed algorithm seems novel in the field, adding new knowledge to the community

Weaknesses:
* The reported results need to be further enhanced, e.g. MCMC and ML classifier design and influence to the final performance were not reported.

**Questions For Authors:**

1. Does the conclusion still hold with different types of classifiers?

2. Did MCMC fail to estimate the correct posterior region in some of the repeated experiments?

**Relation To Broader Scientific Literature:**

This work seems contribute to the consensus prediction in a human-AI using a Bayesian method, highly related to the broader literature in this field, but with a niche problem setting.

**Theoretical Claims:**

Problem setup and Algorithm 1 seem fine.

---

### Official Review · Reviewer_DV8P · 2025-03-14

**Overall Recommendation:** 3

**Summary:**

This paper addresses an interesting problem of predicting human consensus among correlated experts and machine learning models. The end goal is an active learning task, where the consensus label must be predicted using as few experts as possible. To this end, a hierarchical Bayesian model is presented to model "agent" correlations (i.e., correlations of both human experts and machine learning models), and an information gain criteria is presented and numerically estimated for active learning. Several experiments are presented which show the proposed method is able to recover the consensus label with fewer expert queries than other baselines.

## Update after rebuttal

I am glad that the authors will incorporate further prior work and more complete exposition in the final paper. While a theoretically simple modification, performing "online" inference would require non-trivial reimplemtation, in my opinion. Overall, my evaluation of the paper has improved, but I maintain my score of 3.

**Claims And Evidence:**

The main claim made by this submission is that the proposed Bayesian model is appropriate and effective at tackling the proposed problem. I believe this claim to be well-founded, both from the modelling perspective (with similarities to many similar approaches and well-motivated modifications) and the empirical perspective (with its experiments).

**Essential References Not Discussed:**

Following the line of Kim \& Ghahramani (which was discussed in the paper), Trick \& Rothkopf present a solution to correlated experts [1]. This is directly relevant to discussions about the use of Dirichlet priors, as their solution is a version of the Dirichlet distribution with explicit correlation structure. Perhaps closer to the current work is that of Pirš \& Štrumbelj, who use an inverse additive logistic transformation [2].

**Experimental Designs Or Analyses:**

The experiments are well-designed, and I reviewed all of the experiments in Section 7 (i.e., Cost vs. Error, Calibration and Error Bounding, Exploration vs. Exploitation, and the "online" (or continual) learning experiment).

**Methods And Evaluation Criteria:**

Yes, I think the proposed methods and evaluation criteria are appropriate. It is difficult to derive good baselines, since the problem formulation itself is novel, but the choices taken seem reasonable.

**Other Comments Or Suggestions:**

I have a few editorial remarks:
1. There is no punctuation after Eq. (6).
2. The marker for footnote 3 should be placed after the period.

As a more stylistic remark,
1.  The content in Appendix A could be made significantly more readable by including some discussion of each step.

**Other Strengths And Weaknesses:**

### Strengths

**Regarding the Setting** I think the proposed setting is very interesting, and to my knowledge novel.

### Weaknesses

**Regarding Presentation of Equations** I think the presentation of some equations (e.g. Eq. (4) and Eq. (5), and throughout Appendix A) may be improved by including (in words) a brief description. For example, Eq. (4) may be read as "the predictive distribution of $y_*$ can be obtained as a nested expectation, where first correlations between experts are considered"

**Regarding Novelty** There have been several methods in the Bayesian combination literature that attempt the modelling of correlated experts (e.g., [1], [2]), and the entropy-based active learning approach is classical. That said, I think this is balanced by the interesting problem setting.

**Questions For Authors:**

1. I would appreciate if the authors could comment more about the computational demand of the proposed method; for example, what is meant by "over the course of several days." This would help clarify the expense of the proposed method, which can be practically relevant.
2. It is stressed several times that the method is an online one, but to my knowledge, the entire posterior is re-sampled from scratch using NUTS for every value of $t$; is this understanding correct?

### References

[1] Trick, Susanne, and Constantin Rothkopf. "Bayesian classifier fusion with an explicit model of correlation." International Conference on Artificial Intelligence and Statistics. PMLR, 2022.
[2] Pirš, Gregor, and Erik Štrumbelj. "Bayesian combination of probabilistic classifiers using multivariate normal mixtures." Journal of Machine Learning Research 20.51 (2019): 1-18.

**Relation To Broader Scientific Literature:**

This paper proposes an interesting setting where correlated experts should be queried in a cost-aware way. I believe the approach is solid, (though perhaps not particularly novel technically), and forms a valuable contribution to the literature of Bayesian combination of expert knowledge.

**Theoretical Claims:**

The main theoretical claims are regarding the derivations of expectation quantities and algorithms. I checked the expectation expressions, which are relatively simple and seem correct. Algorithm 1 seems correct, though is potentially lacking some details (e.g., in line 3: draw from what posterior?). Algorithm 2 is a straightforward way to approximate the expected entropy using Monte Carlo samples.

---

### Decision · Program_Chairs · 2025-05-01

**Decision:**

Accept (poster)

**Comment:**

The authors introduced a rather interesting method to make predictions with the input of human annotators and trained classifiers by explicitly modeling the correlation between the human experts and the classifier. In general, the reviewers were positive about the paper and the authors provided a suitable rebuttal to the critiques. Therefore I vote to accept the paper.